# Learning a Reusable Meta Denoiser for Learning with Noisy Labels on Multiple Target Domains

## Abstract

*Learning with noisy labels* (LNL) is a classification problem, where some training data are mislabeled. To identify which data are mislabeled, many denoising (e.g., sample selection) methods have been proposed by exploring *meta features* (e.g., loss values) during training. They are successful, since the used meta features are informative for identifying mislabeled data. However, the useful meta features are discarded after training, which is a waste of resources if LNL is needed on more datasets. In this paper, we work on LNL with one clean source domain and multiple noisy target domains and propose a general framework called *meta denoising* (MeDe), where the input spaces and/or label sets can be different for the source and target domains. Specifically, we find that some meta features are nearly *transferable* across datasets; thus, we train a *reusable* meta denoiser, which is a binary classifier to identify mislabeled data given meta features, by simulating noisy labels on the source domain; then, we can run the meta denoiser on any target domain by extracting its own meta features. Experiments show that MeDe can denoise datasets with different label sets and outperform denoising methods applied on each dataset separately.

## 1 Introduction

Deep Neural Networks (DNNs) have achieved great success in many real-world tasks, such as image classification (Russakovsky et al., 2015), information retrieval (Pang et al., 2017), and natural language processing (Devlin et al., 2018). Unfortunately, training DNNs often requires a large number of labeled examples, which are difficult and costly to collect in practice. In order to mitigate this issue, an alternative solution is to adopt some cost-effective strategies such as crowdsourcing (Welinder et al., 2010) for improving labeling efficiency. While the labeling cost is significantly reduced, the learning task becomes much more challenging due to the inevitable noisy labels introduced by such methods. Existing works have validated that DNNs often suffer from the over-fitting issue to label noise, making them fail to obtain favorable generalization performance (Patrini et al., 2017; Han et al., 2018).

Recently, a large number of methods have been developed to train DNNs robustly with noisy labels. The most commonly used strategy is to select examples so as to alleviate the harmfulness of label noise. Among them, early works Han et al. (2018); Yu et al. (2019); Wei et al. (2020) aimed to filter mislabeled examples based on their loss values Jiang et al. (2018). To boost the denoising performance, Li et al. (2020) employed a Gaussian Mixture Model (GMM) based on the losses to divide the training data into a labeled set and an unlabeled set. Then, the well-established semi-supervised learning technique (Berthelot et al., 2019) can be used to train DNNs based on labeled and unlabeled data. To ensure the class-balancing among the selected clean examples, Karim et al. (2022) utilized the Jensen-Shannon divergence (JSD) as the selection criterion.

Thanks to the strong distinctiveness of used meta features, *e.g.*, loss values, these methods have shown improvement in the practical performance for learning with noisy labels. However, once the LNL task has been finished, these methods discard the useful meta features immediately, leading to a huge waste of resources, especially for the case that the LNL is needed on more datasets. Although these meta features cannot be directly applied to other corrupted datasets, they still contain the

common knowledge of label denoising. Therefore, it would be an important problem to improve the model robustness on target domains by exploiting the knowledge of meta features from the source domain.

To tackle the challenge, in this paper, we propose a new learning framework to perform label denoising for noise-corrupted datasets in a meta-learning fashion. Our main idea is to train a reusable meta denoiser based on data-independent meta features that can generalize the transferability of label denoising across datasets. The only need is a clean source dataset that can be easily collected in many real-world tasks (Venkateswara et al., 2017). The proposed Meta Denoising (MeDe) framework consists of two main components, including the meta extractor and meta denoiser. The meta extractor aims to extract meta features that can capture distinctive patterns for distinguishing between mislabeled and clean examples from both source and target domains. This enables us to train a meta denoiser, *i.e.*, a binary classifier, based on source meta data, such that it can generalize well to diverse target domains based on their meta features, even those with different feature spaces and label sets. Finally, the target classifier can be trained with clean training examples identified by the meta denoiser. Theoretical results show that the generalization performance of MeDe depends on the divergence between source and target domains, which can be significantly reduced by the meta extractors.

## 2 RELATED WORK

A large number of methods have been proposed to deal with label noise, and these methods can be roughly divided into three groups, including sample selection (re-weighting), loss robustification, and robust regularization.

The most relevant studies to our work are sample selection methods, which aim to alleviate harmfulness of noisy labels by identifying clean examples. For example, MentorNet (Jiang et al., 2018) first pre-trains an extra network, and then uses it to select clean examples for guiding the training of the classifier network. However, it often suffers from the issue of accumulated error during the training phase. To deal with this issue, Co-teaching (Han et al., 2018) and its variants (Yu et al., 2019; Wei et al., 2020) train two neural networks to avoid errors accumulated on a single network. DivideMix (Li et al., 2020) first incorporates semi-supervised learning techniques into model training. UniCon (Karim et al., 2022) first performs sample selection for corrupted data with the assistance of self-supervised learning techniques. Similar to sample selection methods, sample re-weighting handles noisy labels by adjusting the weights of losses. The popular methods often estimate the weights of losses based on a clean validation set (Ren et al., 2018; Shu et al., 2019).

Robust loss methods design loss functions that are immune to label noise. The early works studied the robustness of some specific loss functions, such as the Mean Absolute Error (MAE) (Ghosh et al., 2017). However, training DNNs with MAE often suffers from the gradient saturation issue, which degrades the model performance. The following work designed the Improved MAE (IMAE) (Wang et al., 2019a) to solve this issue. Some other works designed robust loss functions based on the most commonly used cross entropy loss, such as the Generalized Cross Entropy (GCE) (Zhang & Sabuncu, 2018) loss and symmetric cross entropy loss (Wang et al., 2019b). Ma et al. (2020) proposed a normalization technique to improve the robustness of any loss function theoretically.

A large number of recent studies validate that robust regularization is able to prevent the model from memorizing noisy labels. These methods can be grouped into two categories, the explicit regularization (Liu et al., 2020; 2022; Menon et al., 2019) and implicit regularization (Lukasik et al., 2020; Zhang et al., 2017). The former incorporates a regularization term into model training, such as the over-parameterization term (Liu et al., 2022), while the latter conducts special processing for improving model robustness, such as MixUp Zhang et al. (2017) and label smoothing Lukasik et al. (2020).

**Discussion with Unsupervised Domain Adaptation (UDA)** Given that the target examples can be mislabeled, a straightforward strategy is to treat the target dataset as unlabeled by discarding all of its labels. The most related setting to such a problem is unsupervised domain adaptation (UDA), which leverages the supervision of a source dataset to train a model that generalizes well to the unlabeled target dataset (Ganin & Lempitsky, 2015; Long et al., 2016). However, UDA methods cannot be directly applied to solve our problem, since it requires the source domain and target domain to share

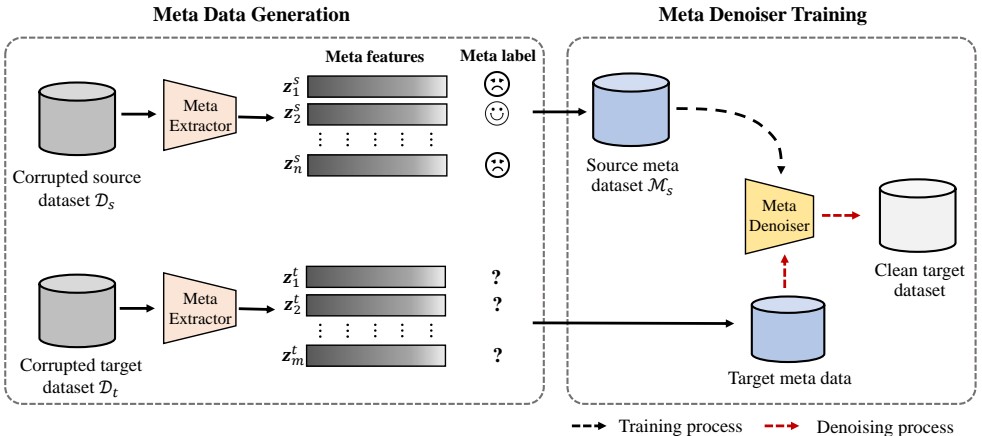

Figure 1: An illustration of the MeDe framework. The framework mainly consists of two components, including the meta extractor, which generates the meta features for corrupted source and target datasets, and the meta denoiser, which generalizes its ability of label denoising to the noisy target dataset by training on the source meta dataset.

the same label space, which hardly holds in our setting. Generally, MeDe assumes that the source domain and target domain share neither the same feature spaces nor the same label space.

## 3 THE MEDE FRAMEWORK

Suppose we are given a source dataset with $n$ examples $\mathcal{D}_s = \{(\boldsymbol{x}_i^s, y_i^s)\}_{i=1}^n$ and a corrupted target dataset with $m$ examples $\widetilde{\mathcal{D}}_t = \{(\boldsymbol{x}_j^t, \tilde{y}_j^t)\}_{j=1}^m$, where $\boldsymbol{x}_i^s$ and $\boldsymbol{x}_j^t$ are the $i$-th and $j$-th instances from the source dataset and target dataset, respectively. Here, we use $y_i^s$ to denote the true label for $\boldsymbol{x}_i^s$, while $\tilde{y}_j^t$ to denote the corrupted label for $\boldsymbol{x}_j^t$. It is noteworthy that we assume that the source dataset and target dataset share neither the same feature space nor the same label space. The goal of MeDe is to train a reusable meta denoiser based on the source dataset $\mathcal{D}_s$ that can generalize to any noisy target dataset $\widetilde{\mathcal{D}}_t$.

The MeDe framework mainly consists of two components, including meta data generation, which produces dataset-independent meta features for source and target training datasets, and meta denoiser training, which trains a binary classifier based on source meta data to predict clean examples for the target domain. Figure 2 provides an illustration of the label denoising process performed by MeDe on one source dataset and one target dataset. It is noteworthy that the trained meta denoiser can be reused on any other target dataset by extracting its own meta features. Firstly, we utilize the meta extractor to generate meta features for the manually corrupted version of the source data. Then, a meta denoiser is trained on the source meta dataset consisting of meta features and binary meta labels that indicate if the examples are mislabeled. Finally, we can run the meta denoiser on the target meta features extracted by the meta extractor to obtain clean target training examples. In the following contents, we will introduce each component of MeDe in detail.

### 3.1 META DATA GENERATION

To train a meta denoiser that can generalize well across datasets, the most important task is to generate dataset-independent meta features for the examples from both source and target domains.

Specifically, we design a meta extractor $\mathcal{E}$ to finish this task. In general, to generate high-quality features for the training of the meta denoiser, the meta extractor $\mathcal{E}$ is expected to satisfy the following two conditions: 1) It can generate distinctive features for separating clean and mislabeled examples. Generally, more distinctive meta features lead to a strong discrimination ability of the meta denoiser, which benefits the final classification model from identifying more clean training

examples. 2) It can generate dataset-independent features for input source and target instances with a small domain shift. This allows a meta denoiser trained on the source meta dataset to generalize well to any target dataset. Generally, we can evaluate the quality of meta features generated by an extractor based on the two aspects, including the distinctiveness $\mathcal{U}_d(\mathcal{E})$ and transferability $\mathcal{U}_t(\mathcal{E})$. The former measures the ability of $\mathcal{E}$ to generate distinctive features that can separate clean and mislabeled examples, while the latter yields ability of $\mathcal{E}$ to generalize across datasets. Based on the aforementioned discussions, we give the formal definition of $\mathcal{E}$ as follows:

**Definition 1.** *Given the source dataset $\mathcal{D}_s$ and target dataset $\mathcal{D}_t$, the meta extractor $\mathcal{E}$ generates meta features $z^s = \mathcal{E}(x^s)$, $z^t = \mathcal{E}(x^t)$ for any $x^s \in \mathcal{D}_s$, $x^t \in \mathcal{D}_t$, with the goal of achieving both high utilities of $\mathcal{U}_d(\mathcal{E})$ and $\mathcal{U}_t(\mathcal{E})$.*

**Discussion**  It is interesting to discuss the specific formulations of these two metrics. The distinctiveness $\mathcal{U}_d(\mathcal{E})$ measures the separation degree of meta features between clean and mislabeled examples. A large distinctiveness $\mathcal{U}_d(\mathcal{E})$ yields that it is easier to separate clean examples from training ones based on the meta features extracted by $\mathcal{E}$. Motivated by recent advances in self-supervised learning (He et al., 2020), in order to evaluate the distinctiveness of a meta extractor $\mathcal{E}$, one can perform linear classification or $k$-NN classification on the generated meta features. For example, we first train a linear classifier on meta features, and then report classification accuracy on the pre-divided test dataset. The transferability $\mathcal{U}_t(\mathcal{E})$ measures the alignment degree between the source and target meta features. The concept is opposite to the divergence (Ganin et al., 2016), which measures the distance between source and target domains.

We provide two options to extract meta features. More elaborate choices can be a direction for future studies. The two options are listed as follows:

- **Loss trajectory**, which records the losses across training epochs of each corrupted training example. Compared with the single training loss value, the loss trajectory captures sufficient distinctive patterns that can distinguish between clean and mislabeled examples. The corresponding method is denoted by MeDe-L.

- **Confidence mismatches**, which record the differences between the maximal confidences and the confidences on the given class across training epochs of each corrupted training example. Compared with the loss trajectory, the confidence mismatch does not only concern model outputs on the given class, but also consider the maximal output among all outputs, which is useful for separating clean and mislabeled examples. The corresponding method is denoted by MeDe-C.

To collect these two kinds of meta features, we need to manually construct a corrupted source dataset $\widetilde{\mathcal{D}}_s = \{(x_i^s, \tilde{y}_i^s)\}_{i=1}^n$ based on the original source dataset $\mathcal{D}_s$. Then, we train a deep neural network denoted by $f(x; \theta_s)$ on the corrupted dataset $\widetilde{\mathcal{D}}_s$ by using the ordinary stochastic gradient descent (SGD) optimizer with the traditional cross entropy (CE) loss:

$$\mathcal{L}(\widetilde{\mathcal{D}}_s, \theta_s) = -\sum_{i=1}^n \sum_{k=1}^K \tilde{y}_i^s \log p_k(x_i^s), \tag{1}$$

where $p_k(x_i^s)$ denotes the $k$-th component of the output $f(x_i^s; \theta_s)$, *i.e.*, the prediction probability of the $k$-th class for instance $x_i^s$. During the training phase, at the $t$ epoch, we record the loss trajectory $z_{it} = -\log p_{\tilde{y}_i}(x_i)$ (or confidence mismatch $z_{it} = \max_k p_k(x_i) - p_{\tilde{y}_i}(x_i)$) for instance $x_i$. After the training process, by arranging the recorded information, we can obtain the source meta dataset $\mathcal{M}_s = \{(z_i^s, v_i^s)\}_{i=1}^n$, where $v_i^s$ is the meta label that indicates if example $x_i^s$ is mislabeled. Here, $v_i^s = 0$ indicates instance $x_i^s$ is mislabeled; $v_i^s = 1$, otherwise. It is noteworthy that we can obtain the meta label $v_i^s$ without any labeling cost, since we assume that the true labels of training examples from the source domain are accessible. Similarly, by training a neural network $f(x; \theta_t)$ on the target dataset, we can obtain the target meta dataset $\mathcal{M}_t = \{z_j^t\}_{j=1}^m$. Our goal is to train a meta denoiser based on source meta dataset $\mathcal{M}_s$ that can accurately predict the true meta label $v_j^t$ for $z_j^t$ from the target domain.

### 3.2 META DENOISER TRAINING

In Section 3.1, we introduced the details of collecting meta datasets. Next, we discuss how to train a meta denoiser on the source meta dataset. Existing methods often perform label denoising based

---

**Algorithm 1** The MeDe Algorithm

---

1: **Input:**
2:   $\mathcal{D}_s$: the source dataset
3:   $\widetilde{\mathcal{D}}_t$: the corrupted target dataset
4: **Process:**
5:   **Meta data generation**
6:   Initialize the network parameters $\theta_s$ and $\theta_t$; construct the corrupted dataset $\widetilde{\mathcal{D}}_s$ based on the source dataset $\mathcal{D}_s$.
7:   Train the network $f(\boldsymbol{x}; \theta_s)$ on the corrupted source dataset $\widetilde{\mathcal{D}}_s$, and obtain the source meta dataset $\mathcal{M}_s = \{(\boldsymbol{z}_i^s, v_i^s)\}_{i=1}^n$.
8:   Train the network $f(\boldsymbol{x}; \theta_t)$ on the corrupted target dataset $\widetilde{\mathcal{D}}_t$, and obtain the target meta dataset $\mathcal{M}_t = \{\boldsymbol{z}_j^t\}_{j=1}^m$.
9:   **Meta denoiser training**
10:   Initialize the parameter $\psi$.
11:   Train the meta denoiser $w(\boldsymbol{z}; \psi)$ on source meta dataset $\mathcal{M}_s$ by utilizing a off-the-shelf supervised learning method.
12:   Use the meta denoiser $w(\boldsymbol{z}; \psi)$ to achieve label denoising for the target meta dataset $\mathcal{M}_t$, and then obtain the clean target dataset $\mathcal{D}_t$.
13: **Output:** The clean target dataset $\mathcal{D}_t$.

---

on single values of meta features, which only contains limited information. In contrast to existing works, we train a meta denoiser on the source feature vectors that encode rich training information. The intuition behind the meta denoiser is that with informative training data, it is able to learn the ability to separate clean and mislabeled examples and generalize such ability across datasets. This yields that the meta denoiser can be used to perform label denoising for the target meta dataset. Specifically, given the source meta dataset $\mathcal{M}_s = \{(\boldsymbol{z}_i^s, v_i^s)\}_{i=1}^n$, we train the meta denoiser denoted by $w(\boldsymbol{z}; \psi)$, where $\psi$ is the corresponding parameter, by utilizing off-the-shelf supervised learning methods, such as a neural network, logistic regression, and the support vector machine (SVM).

After obtaining the meta denoiser, we can use it to perform label denoising for the target meta dataset. Specifically, we first predict the probability $p(\boldsymbol{z}_j^t) = w(\boldsymbol{z}_j^t; \psi)$ for all target meta data $\boldsymbol{z}_j^t \in \mathcal{M}_t$. Next, we rank all target meta data according to their prediction probabilities in a descending order, and then select top $\tau\%$ rank examples as clean ones. The main procedures of the MeDe method are summarized in Algorithm 1. Finally, we train the target classifier based on the identified clean examples in a supervised learning manner. Alternatively, to further improve the classification performance, similar to previous works, we can train the target classifier on both the identified clean examples and the remaining ones by either using semi-supervised learning techniques (Wang et al., 2020; Li et al., 2020; Bai et al., 2021) or performing *stochastic gradient ascent* on unselected data (Han et al., 2020).

## 4 THEORETICAL ANALYSIS

After collecting clean training examples, we train a neural network denoted by $f(\boldsymbol{x}; \theta_c)$ to perform target classification based on the following weighted CE loss:

$$\mathcal{L}(\tilde{D}_t, \theta_c) = -\sum_{j=1}^m w(\boldsymbol{z}_j^t; \hat{\psi}) \log p_{\tilde{y}_j^t}(\boldsymbol{x}_j^t), \tag{2}$$

where $p_{\tilde{y}_j^t}(\boldsymbol{x}_j^t)$ denotes the $\tilde{y}_j^t$-th component of $f(\boldsymbol{x}_j^t; \theta_t)$, and $w(\boldsymbol{z}_j^t; \hat{\psi}) \in \{0, 1\}$ is the weight for the $j$-th target example predicted by the meta denoiser $\hat{\psi}$ trained on the source meta dataset. Here, $w(\boldsymbol{z}_j^t; \hat{\psi}) = 1$ indicates the $j$-th example is selected for target classifier training, while $w(\boldsymbol{z}_j^t; \hat{\psi}) = 0$, otherwise. From Eq.equation 2, it is obvious that the performance of the target classifier highly depends on the quality of weights. If the predicted weights are more accurate, then the target classifier achieves better performance. In the following contents, we study the generalization performance of the meta denoiser from the theoretical perspective.

To quantify the performance of the meta denoiser, we define its risk with respect to the source distribution as

$$R_s(\psi) = \mathbb{E}[\ell(w(\boldsymbol{z}^s; \psi), v^s)], \tag{3}$$

and the target distribution as

$$R_t(\psi) = \mathbb{E}[\ell(w(\boldsymbol{z}^t; \psi), v^t)], \tag{4}$$

where $\ell$ is a loss function. Compared with the source risk $R_s(\psi)$, we pay more attention to the target risk $R_t(\psi)$, since our goal is to predict accurate weights for target training examples. Furthermore, we define the empirical version of the source risk as

$$\widehat{R}_s(\psi) = \frac{1}{n} \sum_{i=1}^{n} \ell(w(\boldsymbol{z}_i^s; \psi), v_i^s), \tag{5}$$

which is the only accessible information during the training process.

We provide a generalization error bound for our proposed MeDe method to show its learning consistency across datasets. Let $\hat{\psi} = \min_{\psi \in \Psi} \widehat{R}_s(\psi)$ be the empirical optimizer that minimizes the empirical source risk $\widehat{R}_s(\psi)$, and $\psi^* = \arg\min_{\psi \in \Psi} R_t(\psi)$ be the true optimizer that minimizes target risk $R_t(\psi)$, where $\Psi$ is a function space. Let $\mathcal{R}_m(\Psi)$ be the expected Rademacher complexity (Mohri et al., 2018) of $\Psi$ with $m$ training points.

**Theorem 1.** *Suppose that the loss function $\ell$ is $L_\ell$-Lipschitz continuous w.r.t. $\psi$ and bounded by $B$. Define the $\mathcal{H}$-divergence between source and target domains as $d_{\mathcal{H}}(p, q)$, where $p$ and $q$ respectively correspond to distributions of source and target domains. For any $\delta > 0$, with the probability at least $1 - \delta$, we have*

$$R_t(\hat{\psi}) \leq R_t(\psi^*) + d_{\mathcal{H}}(p, q) + 4L_\ell \mathcal{R}_m(\Psi) + 2B\sqrt{\frac{\log 1/\delta}{2n}}.$$

Theorem 1 tells us that the generalization performance of MeDe is dependent on two elements. The first one is the domain shift between source and target domains, which can be measured by the divergence $d_{\mathcal{H}}(p, q)$ (Ben-David et al., 2006). Generally, smaller $\mathcal{H}$-divergence leads to better generalization performance. Although it is very hard to quantify the divergence, we provide a qualitative analysis for the domain shift between source and target meta features based on their visualization results in Section 5.4. Our empirical findings show that by utilizing the meta extractor, the domain shift between source and target meta features can be very small. The second element is the number of source training examples $n$. As $n \to \infty$, by neglecting the domain divergence, we can achieve the learning consistency: $R_t(\hat{\psi}) \to R_t(\psi^*)$.

## 5 EXPERIMENTS

In this section, we perform experiments on multiple benchmark and realistic datasets to validate the effectiveness of the proposed method.

### 5.1 EXPERIMENTAL SETTING

**Datasets**  We conduct experiments to evaluate our proposed method on three benchmark datasets: CIFAR-10, CIFAR-100 [1] (Krizhevsky et al., 2009), and STL-10 [2] (Coates et al., 2011), as well as two real-world corrupted datasets: CIFAR10-N and CIFAR100-N [3] (Wei et al., 2022). These datasets have been widely used for the evaluation of learning with noisy labels in the previous literature (Karim et al., 2022; Liu et al., 2022). Except for the two realistic corrupted datasets, following the previous works (Reed et al., 2014; Patrini et al., 2017), we construct the corrupted version of remaining datasets by using the label transition matrix $Q$, where $Q_{jk} = Pr(\tilde{y} = k|y = j)$ represents the probability of label $y$ to be flipped into label $\tilde{y}$. In our experiments, we assume that there exist two types of label transition matrices: 1) Symmetric flipping Van Rooyen et al. (2015): each class label being flipped into one of rest class labels uniformly. 2) Asymmetric flipping (Wei et al., 2020):

---

[1]`https://www.cs.toronto.edu/˜kriz/cifar.html`
[2]`https://cs.stanford.edu/˜acoates/stl10`
[3]`http://www.noisylabels.com`

Table 1: Comparison results on CIFAR-10 and CIFAR-100 in terms of accuracy (%) with {20%,40%.50%} percent of symmetric and {10%,30%,40%} percent of asymmetric noise. The best performance is highlighted in bold.

| Method | CIFAR-100 → CIFAR-10 | | | | | | CIFAR-10 → CIFAR-100 | | | | | |
| | Symmetric | | | Asymmetric | | | Symmetric | | | Asymmetric | | |
| | 20% | 40% | 50% | 10% | 30% | 40% | 20% | 40% | 50% | 10% | 30% | 40% |
|---|---|---|---|---|---|---|---|---|---|---|---|---|
| CE | 86.8 | 81.7 | 79.4 | 88.8 | 81.7 | 76.1 | 62.0 | 51.2 | 46.7 | 68.1 | 53.3 | 44.5 |
| LDMI | 88.3 | - | 81.2 | 91.1 | 91.2 | 84.0 | 58.8 | - | 51.8 | 68.1 | 54.1 | 46.2 |
| PENCIL | 92.4 | - | 89.1 | 93.1 | 92.9 | 91.6 | 69.4 | - | 57.5 | 76.0 | 59.3 | 48.3 |
| JNPL | 93.5 | 91.9 | 90.2 | 94.2 | 92.5 | 90.7 | 70.9 | 68.1 | 67.7 | 72.0 | 68.1 | 59.5 |
| MOIT | 94.1 | 91.2 | 91.1 | 94.2 | 94.1 | 93.2 | 75.9 | 70.9 | 70.1 | 77.4 | 75.1 | 74.0 |
| DivideMix | 96.1 | 94.9 | 94.6 | 93.8 | 92.5 | 91.7 | 77.3 | 75.9 | 74.6 | 71.6 | 69.5 | 55.1 |
| ELR | 95.8 | 95.1 | 94.8 | 95.4 | 94.7 | 93.0 | 77.6 | 75.2 | 73.6 | 77.3 | 74.6 | 73.2 |
| UniCon | 96.0 | 95.6 | 95.6 | 95.3 | **94.8** | **94.1** | 78.9 | 78.1 | 77.6 | 78.2 | 75.6 | 74.8 |
| MeDe-L | 96.8 | **96.6** | **95.7** | 95.4 | 92.4 | 91.0 | **80.8** | **78.7** | **77.7** | 80.3 | 75.9 | 68.4 |
| MeDe-C | **96.9** | 96.1 | 95.5 | **95.5** | 92.7 | 91.5 | 80.2 | 78.2 | 76.5 | **80.8** | **79.7** | **77.7** |

a simulation of fine-grained classification with noisy labels, where labelers may mislabel only within very similar classes.

**Comparing methods** To validate the effectiveness of our proposed MeDe-L and MeDe-C, we compare them with the following state-of-the-art algorithms: **UniCon** (Karim et al., 2022), which incorporated contrastive learning techniques into the sample selection method; **SOP** (Liu et al., 2022), which trained DNNs with a sparse over-parameterization term that models the label noise; **PES** (Bai et al., 2021), which designed a progressive early stopping training strategy to prevent the latter DNN layers from being over-fitted to noisy labels; **ELR** (Liu et al., 2020), which designed a regularization term to implicitly prevent memorization of noisy labels; **DivideMix** (Li et al., 2020), which leveraged semi-supervised learning techniques for improving robustness against label noise. Full information of all comparing methods can be found in the appendix.

**Implementation** Following the previous works (Liu et al., 2022; Bai et al., 2021), for all datasets, we use PreActResNet18 architecture. At the stage of meta feature generation, we train networks on the corrupted source dataset and corrupted target dataset with SGD optimizer. We run 300 epochs in total with the initial learning rate of 0.02, and use the cosine annealing (Loshchilov & Hutter, 2017) scheduler to adjust the learning rate during the training process. At the stage of meta denoiser training, to reduce computational complexity, we train a logistic regression on the generated source meta dataset by adopting the implementation of Scikit-learn (Pedregosa et al., 2011), and use it to perform denoising for different target datasets. To further improve the performance of the final classifier, similar to previous works (Li et al., 2020; Bai et al., 2021), we employ semi-supervised learning techniques (Berthelot et al., 2019) to train the neural networks on both identified clean examples and remaining mislabeled ones. The random seed is set to 1 for all experiments. We perform all experiments on GeForce RTX 2080 GPUs.

## 5.2 RESULTS ON MULTIPLE TARGET DOMAINS

In this section, we first report the results on CIFAR-10 and CIFAR-100 to validate the denoising ability of MeDe, and then report the results on STL-10 to verify its ability to generalize across multiple target domains.

Table 1 reports the test accuracy of each comparing method on CIFAR-10 and CIFAR-100 with symmetric and asymmetric noise. For MeDe, we use one of CIFAR-10 and CIFAR-100 as the source dataset, while the other as the target dataset; for the comparing methods, we directly train models on the target dataset. From the table, it can be observed that MeDe achieves the best performance on almost all cases excepted for CIFAR-10 with 30% and 40% asymmetric noise, where UniCon achieves the best performance. It is noteworthy that even compared with the SOTA method UniCon that adopts contrastive learning techniques, our method achieves competitive performance and shows better performance than UniCon in most cases. Furthermore, compared with MeDe-L, MeDe-C obtains better performance, especially on CIFAR100. This is because MeDe-C extracts

Table 2: Comparison results on STL-10 in terms of accuracy (%) with {20%,40%,50%,60%} percent of symmetric noise and {10%,30%,40%,50%} percent of asymmetric noise. The best performance is highlighted in bold.

| Method | CIFAR-10 → STL-10 | | | | | | | |
|---|---|---|---|---|---|---|---|---|
| | Symmetric | | | | Asymmetric | | | |
| | 20% | 40% | 50% | 60% | 10% | 30% | 40% | 50% |
| MOIT | 61.48 | 61.25 | 58.63 | 52.16 | 59.64 | 59.43 | 58.74 | 58.26 |
| SOP | 64.90 | 58.20 | 51.05 | 38.30 | 68.54 | 56.60 | 49.28 | 46.65 |
| PES | 66.28 | 68.77 | 62.10 | 56.72 | 64.17 | 63.09 | 55.27 | 52.93 |
| DivideMix | 72.13 | 74.23 | 74.20 | 72.84 | 66.83 | 68.84 | 65.04 | 60.80 |
| Unicon | 74.60 | 74.53 | 72.75 | 71.03 | 71.74 | 71.03 | 69.75 | 68.20 |
| MeDe-L | **84.20** | **80.28** | 75.25 | 70.11 | 85.50 | 80.63 | 77.80 | **73.83** |
| MeDe-C | 83.04 | 79.56 | **76.41** | **72.87** | **85.57** | **82.01** | **78.96** | 73.52 |

Table 3: Comparison results on real-world corrupted datasets CIFAR-N. Mean and standard deviation over 3 independent runs are reported. The results of comparing methods are copied from the official leaderboard Wei et al. (2022). The best performance is highlighted in bold.

| Method | CIFAR-100N→CIFAR-10N | | | | | CIFAR-100N |
|---|---|---|---|---|---|---|
| | Aggre | Random1 | Random2 | Random3 | Worst | Noisy |
| CE | $87.77_{\pm 0.38}$ | $85.02_{\pm 0.65}$ | $86.46_{\pm 1.79}$ | $85.16_{\pm 0.61}$ | $77.69_{\pm 1.55}$ | $55.50_{\pm 0.66}$ |
| ELR | $91.09_{\pm 0.10}$ | $94.43_{\pm 0.41}$ | $94.20_{\pm 0.24}$ | $94.34_{\pm 0.22}$ | $94.83_{\pm 1.60}$ | $66.72_{\pm 0.07}$ |
| CORES | $95.25_{\pm 0.09}$ | $94.45_{\pm 0.14}$ | $94.88_{\pm 0.31}$ | $94.74_{\pm 0.03}$ | $91.66_{\pm 0.09}$ | $61.15_{\pm 0.73}$ |
| Divide-Mix | $95.01_{\pm 0.71}$ | $95.16_{\pm 0.19}$ | $95.23_{\pm 0.07}$ | $95.21_{\pm 0.14}$ | $92.56_{\pm 0.42}$ | $71.13_{\pm 0.48}$ |
| PES | $94.66_{\pm 0.18}$ | $95.06_{\pm 0.15}$ | $95.19_{\pm 0.23}$ | $95.22_{\pm 0.13}$ | $92.68_{\pm 0.22}$ | $70.36_{\pm 0.33}$ |
| SOP | $95.61_{\pm 0.13}$ | $95.28_{\pm 0.13}$ | $95.31_{\pm 0.10}$ | $95.39_{\pm 0.11}$ | $93.24_{\pm 0.21}$ | $67.81_{\pm 0.23}$ |
| MeDe-L | $\mathbf{95.70}_{\pm 0.05}$ | $\mathbf{95.88}_{\pm 0.14}$ | $95.99_{\pm 0.13}$ | $95.89_{\pm 0.27}$ | $\mathbf{94.95}_{\pm 0.05}$ | $71.45_{\pm 0.38}$ |
| MeDe-C | $95.70_{\pm 0.09}$ | $95.86_{\pm 0.06}$ | $\mathbf{96.11}_{\pm 0.06}$ | $\mathbf{95.91}_{\pm 0.35}$ | $94.89_{\pm 0.12}$ | $\mathbf{71.91}_{\pm 0.21}$ |

meta features based on two kinds of information, *i.e.*, the maximal output and the output on the given class, while MeDe-L only considers the latter. These results convincingly validate the strong denoising ability of MeDe.

To further validate its reusability to multiple target domains, we apply the meta denoiser trained on CIFAR-10 to STL-10. It is noteworthy that STL-10 and CIFAR-10 have both different feature spaces and label sets, making the task become much more challenging. Table 2 reports the test accuracy of each comparing method on STL10 with symmetric and asymmetric noise. MeDe achieves the best performance in all cases and significantly outperforms the comparing methods. Some methods obtain unfavorable performance mainly due to the following two reasons: 1) Compared with CIFAR-10, the number of training examples on CIFAR-10 is very small; 2) It is hard to tune the parameters on STL-10, since their default parameters are set for CIFAR-10. These results validate that MeDe can effectively generalize its ability across multiple target domains.

### 5.3 RESULTS ON REAL-WORLD DATASETS

To validate the practical usefulness of the proposed method, we perform experiments on realistic corrupted datasets CIFAR-10N and CIFAR-100N. The labels of these two datasets are annotated by three independent labelers from Amazon Mechanical Turk (Wei et al., 2022). For CIFAR-100, since it only has a noisy label set "Noisy", we use CIFAR-10N with the noisy label set "Worst" as the source dataset. For CIFAR-10N, to further validate the reusability of MeDe to multiple target domains in realistic scenarios, regarding different noisy label sets, we use CIFAR-100N with the noisy label set "Noisy" as the source dataset.

Table 3 reports the test accuracy of each comparing method on CIFAR-10N and CIFAR-100N. From the table, we can see that: 1) MeDe obtains desirable performance and significantly outperforms comparing methods in all cases. This discloses that MeDe can achieve effective label denoising

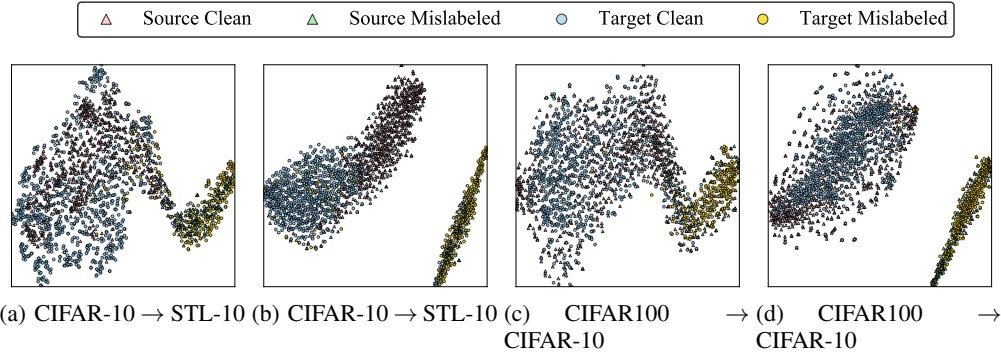

(a) CIFAR-10 → STL-10 (b) CIFAR-10 → STL-10 (c) CIFAR100 → (d) CIFAR100 →
CIFAR-10 CIFAR-10

Figure 2: The visualization results of meta features by utilizing the t-SNE technique. Specifically, (a) and (b) illustrate the visualization results of meta features extracted by MeDe-L on CIFAR-10 and STL-10 at the 50-th and the 300-th epoch. Similarly, (c) and (d) illustrate the visualization results of meta features extracted by MeDe-C on CIFAR-100 and CIFAR-10 at the 50-th and the 300-th epoch. From the figures, it can be observed that the divergence between source and target domains is significantly reduced by the proposed meta extractors.

on realistic corrupted datasets. 2) MeDe-L and MeDe-C show comparable performance, which validates that both two meta extractors can produce high-quality meta features for the sequential meta denoiser training. 3) Based a single source dataset, MeDe consistently achieves favorable performance across diverse target datasets with different label sets. This demonstrates that MeDe can generalize the ability of label denoising across multiple target datasets effectively. The practical usefulness of the proposed method can be sufficiently validated by these experimental results.

## 5.4 VISUALIZATION OF META FEATURES

In section 3.1, we discuss how to quantify the quality of meta features. In this section, to further study the mechanism behind MeDe, we visualize two kinds of meta features by utilizing t-SNE technique (Van der Maaten & Hinton, 2008). In this experiment, Figure 2(a) and Figure 2(b) illustrate the visualization results of meta features generated by MeDe-L on CIFAR-10 (source) and STL-10 (target). Figure 2(c) and Figure 2(d) illustrate the visualization results of meta features generated by MeDe-C on CIFAR-100 (source) and CIFAR-10 (target).

From the figures, it can be observed that: 1) At the begin of model training, there is even no distribution shift between source and target domains. 2) At the end of training, MeDe-L has a larger distribution shift between source and target domains, while MeDe-C still maintains consistent distribution. 3) At the end of training, both MeDe-L and MeDe-C are able to distinguish between clean and mislabeled examples effectively. These visualization results qualitatively validate that the meta features are: 1)distinctive for separating clean and mislabeled examples; 2) transferable between source and target datasets.

## 6 CONCLUSION

In this paper, we study the problem of learning with noisy labels on one clean source dataset and multiple noisy target datasets. Unlike existing methods that discard the useful meta features once the LNL task has been finished, we aim to perform label denoising based on the clean source dataset in a meta-learning fashion. Specifically, we train a reusable meta denoiser based on two kinds of nearly dataset-independent meta features, *i.e.*, the loss trajectory and confidence mismatches, and generalize its ability to multiple target datasets. Based on the informative meta features, the meta denoiser can obtain strong discrimination ability to separate clean examples and mislabeled ones. Comprehensive experiments validate the proposed method can achieve state-of-the-art performance. In the future, we plan to apply MeDe to more realistic tasks.

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

## A  Proof of Theorem 1

Before proving the theorem, we first provide the following two lemmas:

**Lemma 1.** *Suppose the loss function $\ell$ is bounded by $B$. For any $\delta > 0$, with probability at least $1 - \delta$,*

$$|R(\psi) - \widehat{R}(\psi)| \leq 2L_\ell \mathcal{R}_m(\Psi) + B\sqrt{\frac{\log 1/\delta}{2n}} \tag{6}$$

In order to prove the lemma, we can prove that one side $\sup_{\psi \in \Psi} R(\psi) - \widehat{R}(\psi)$ is bounded with probability no less $1 - \delta$, which is a simple extension of Theorem 3.3 in (Mohri et al., 2018). The other side $\widehat{R}(\psi) - \sup_{\psi \in \Psi} R(\psi)$ can be bounded in a similar manner.  □

**Lemma 2.** *For any $\psi \in \Psi$,*

$$|R_t(\psi) - R_s(\psi)| \leq \frac{1}{2}d_\mathcal{H}(p, q). \tag{7}$$

In order to prove the lemma, we need to prove two directions. The first direction $R_t(\psi) \leq R_s(\psi) + \frac{1}{2}d_\mathcal{H}(p, q)$ has been proved by Ben-David et al. (2006). Below, we prove the other direction $R_s(\psi) \leq R_t(\psi) + d_\mathcal{H}(p, q)$.

Let $\psi' = \arg\min_{\psi \in \Psi} R_s(\psi) + R_t(\psi)$ be the true minimizer of the joint risk and $\gamma$ is the corresponding risk $\gamma = R_s(\psi') + R_t(\psi')$. Then, we have:

$$
\begin{aligned}
R_s(\psi) &\leq R_s(\psi') + R_s(\psi, \psi') \\
&\leq R_s(\psi') + R_t(\psi, \psi') + |R_s(\psi, \psi') - R_t(\psi, \psi')| \\
&\leq R_s(\psi') + R_t(\psi, \psi') + \frac{1}{2}d_\mathcal{H}(p, q) \\
&\leq R_s(\psi') + R_t(\psi) + R_t(\psi') + \frac{1}{2}d_\mathcal{H}(p, q) \\
&= R_t(\psi) + \frac{1}{2}d_\mathcal{H}(p, q) + \gamma
\end{aligned}
\tag{8}
$$

where the second line is based on Lemma 3 of (Ben-David et al., 2010).  □

Based on the above two lemmas, for any $\delta > 0$, with probability no less than $1 - \delta$, we have:

$$
\begin{aligned}
R_t(\hat{\psi}) &\leq R_s(\hat{\psi}) + \frac{1}{2}d_\mathcal{H}(p, q) \\
&\leq \widehat{R}_s(\hat{\psi}) + \frac{1}{2}d_\mathcal{H}(p, q) + 2L_\ell \mathcal{R}(\Psi) + B\sqrt{\frac{\log 1/\delta}{2n}} \\
&\leq \widehat{R}_s(\psi) + \frac{1}{2}d_\mathcal{H}(p, q) + 2L_\ell \mathcal{R}(\Psi) + B\sqrt{\frac{\log 1/\delta}{2n}} \\
&\leq R_s(\psi) + \frac{1}{2}d_\mathcal{H}(p, q) + 4L_\ell \mathcal{R}(\Psi) + 2B\sqrt{\frac{\log 1/\delta}{2n}} \\
&\leq R_t(\psi) + d_\mathcal{H}(p, q) + 4L_\ell \mathcal{R}(\Psi) + 2B\sqrt{\frac{\log 1/\delta}{2n}}
\end{aligned}
$$

The first and fifth line are based on Eq.equation 7. The second line and fourth line are due to the bound in Eq.equation 6. The third line is by the definition of $\hat{\psi}$. By substituting $\psi$ with $\psi^*$, we can get the final results.  □

## B  More Experimental Settings

Besides the methods mentioned in the main paper, we also compare the proposed method with the following algorithms: **LDMI** (Xu et al., 2019), which trains DNNs based on a information-theoretical loss function; **PENCIL**, which simultaneously updates the net parameter and true label distribution; **JNPL** (Kim et al., 2021), which jointly performs negative and positive learning to train robust DNNs; **MOIT** (Ortego et al., 2021), which utilizes contrastive learning techniques to exploits the robust feature representations. **CORES** (Cheng et al., 2021), which trains robust DNNs by progressively sieving out corrupted examples.

