# OpenReview forum: "Learning a Reusable Meta Denoiser for Learning with Noisy Labels on Multiple Target Domains"
_ICLR.cc/2024/Conference — Submitted to ICLR 2024_

### Official Review · Reviewer_WBJ3 · 2023-10-30

**Soundness:** 3 good
**Presentation:** 3 good
**Contribution:** 2 fair
**Rating:** 6
**Confidence:** 3

**Summary:**

This paper tries to learn a reusable meta denoise to handle the noisy labels on multi-target learning. It designs two methods to extract the meta features, i.e., MeDe-L and MeDe-C. The paper also conducts some experiments to show  the effectiveness.

**Strengths:**

1.  The motivation of the reusable meta denoiser makes sense.
2.  The paper proposes two methods to extract the meta features.
3.  The overall framework is simple and easy to follow.

**Weaknesses:**

1. It would be better to clarify the contributions more clearly in Introduction.
2. How does the experiments conduct? Since the experiments involves random noises, it should be repeated several times. Are the experiments repeated? If so, in Tables 1 and 2, both the average results and the standard deviation should be reported. Especially, in Table 1, in some cases, the proposed methods perform very closely to some baseline methods.
3. The experimental results show that the performance of MeDe-L is very close to MeDe-C in most cases. If so, what is the necessaty for proposing these two methods?
4. On the real-world data sets, i.e., Table 3, the average reports and standard deviation are reported. However, the difference between the proposed ones and the compared baselines is so small. It seems that the difference is not statistically significant.

**Questions:**

See above.

---

### Official Review · Reviewer_RFE6 · 2023-10-30

**Soundness:** 2 fair
**Presentation:** 2 fair
**Contribution:** 2 fair
**Rating:** 3
**Confidence:** 3

**Summary:**

This article focuses on the problem of learning with noisy labels across multiple target domain. To address this problem, this article proposes an approach based on the idea of Meta Learning. Experiments have been conducted to validate its effectiveness.

**Strengths:**

This work proposes an interesting setting for the noisy label problem and it is also reasonable and natural to introduce Meta Learning to address it. This might provide some inspirations for related studies.

**Weaknesses:**

1)	This article has very limited innovation; it combines ideas from various fields such as meta-learning, learning with noisy labels, and domain adaptation, but it does not propose any meaningful techniques to integrate them effectively. Furthermore, the article is quite vague in its presentation. In the third section, it provides only a rough framework of the proposed method and lacks necessary discussions on algorithm details and potential issues that may arise. For example, it is not clear how the corrupted source data are generated in the proposed method.
2)	In the training phase, this article directly trains a Denoiser on the noisy source data and applies it for the target data. The effectiveness of such process needs to be based on the following premises: First, there should exist some learnable connections between the source data and target data. Second, the user should be aware of the form of label noise in the target data and be able to artificially generate similar noise on the source data. However, the proposed method does not involve any domain adaptation aspects, nor does it make any assumptions about label noise. Besides, it also does not make much sense to extract features from two different and independent neural networks while apply only one denoiser.
3)	The theoretical analysis in the article is directly derived from the classical generalization analysis based on the Rademacher complexity. It is unclear how this theory is related to the noisy label problem studied in the article, and it does not provide any assistance in theoretically understanding the proposed algorithm.

**Questions:**

1)	This article has very limited innovation; it combines ideas from various fields such as meta-learning, learning with noisy labels, and domain adaptation, but it does not propose any meaningful techniques to integrate them effectively. Furthermore, the article is quite vague in its presentation. In the third section, it provides only a rough framework of the proposed method and lacks necessary discussions on algorithm details and potential issues that may arise. For example, it is not clear how the corrupted source data are generated in the proposed method.
2)	In the training phase, this article directly trains a Denoiser on the noisy source data and applies it for the target data. The effectiveness of such process needs to be based on the following premises: First, there should exist some learnable connections between the source data and target data. Second, the user should be aware of the form of label noise in the target data and be able to artificially generate similar noise on the source data. However, the proposed method does not involve any domain adaptation aspects, nor does it make any assumptions about label noise. Besides, it also does not make much sense to extract features from two different and independent neural networks while apply only one denoiser.
3)	The theoretical analysis in the article is directly derived from the classical generalization analysis based on the Rademacher complexity. It is unclear how this theory is related to the noisy label problem studied in the article, and it does not provide any assistance in theoretically understanding the proposed algorithm.

---

### Official Review · Reviewer_Gghz · 2023-10-31

**Soundness:** 3 good
**Presentation:** 3 good
**Contribution:** 2 fair
**Rating:** 5
**Confidence:** 3

**Summary:**

The paper describes an interesting way of filtering data instances with noisy labels across domains by using meta features learned from source domain that generalize over a target domain where source domain may have noisy labels. Particularly, two feature extraction processes are evaluated: one using losses over training epochs on noisy labeled data, and another using confidence during training. By using these features, the authors show that building a binary classifier over these features can effectively utilize good labeled data in the source domain over a related target domain. Their empirical evaluation shows that the proposed framework performs well on multiple datasets in comparison with competing methodologies.

**Strengths:**

1. The solution to the learning with noisy labels provided in the paper seems unique and intriguing. The authors have described the motivation, description and evaluation well.
2. The theoretical analysis demonstrates how source and target domain labels and patterns need to be related for the solution to work.
3. The empirical analysis shows good performance on domain transferability and generalizability of the meta features obtained.

**Weaknesses:**

1. The use of binary classifier over meta features seems to be an excellent idea. However, the paper mentions that the top T (tau) data instances from the final probability list of target instances are considered. What is this threshold used in the evaluation? An ablation study regarding this threshold would be good to add. Furthermore, does this threshold depend on the performance of the binary classifier? These set of questions needs to be addressed and is completely missing in the paper.
2. The paper mentions that the authors tried multiple classifiers such as SVM, neural network etc. What worked and what did not work? Which specific classifier was used to obtain the results presented in Table 2, 3 and 4? What were their performance on the source domain used?
3. Since the solution assumes that the meta-label, i.e., whether the given class is noisy or not, is available, is this not additional effort over the source data labeling effort to create such meta-labels? Also, do you assume that these meta-labels are noise-free? What would happen if this meta-label itself is noisy?
4. As table 2 shows accuracy on symmetric and asymmetric label flipping, the accuracy scores across various competing methods seems close to each other. Is the difference statistically significant?

**Questions:**

What is this threshold Tau used in the evaluation?
Which specific classifier was used to obtain the results presented in Table 2, 3 and 4? And what were their performance on the source domain used?
In Table 3, why is UniCon not mentioned as this competing method seems to be closest to the proposed solution, and best performing as shown in Table 2?
Are the results in Table 2 statistically significant?

---

### Official Review · Reviewer_WbSU · 2023-11-04

**Soundness:** 2 fair
**Presentation:** 2 fair
**Contribution:** 2 fair
**Rating:** 3
**Confidence:** 4

**Summary:**

The paper proposes a method to identify labeling errors using meta-features such as losses and confidence of trained models. These features are independent of a dataset. Given a source dataset (I am assuming without any labeling errors), MeDe first adds label noise to the dataset, and then trains a model to predict label errors solely based on embeddings of meta-features. These embeddings are similarly derived for target dataset, and the same model from the source dataset, is used to identify labeling errors on the target dataset.

**Strengths:**

1. I think the paper proposes an interesting idea under the premise that meta-features such as training loss dynamics of a model and the confidence of a trained models, transfer across datasets.
2. The paper has some theoretical insight on why there method works.

**Weaknesses:**

I believe that the authors should work on the following to improve their work:
1. **Experimentation**: There are several limitations in terms of the experiments that authors have conducted. I would encourage them to go through the recently proposed AQuA benchmark [1] to make their experimentation much more rigorous. Some of these limitations in decreasing order of importance are:
(1) *Recent baselines*: MeDe is not compare with many recent state-of-the-art baselines such as AUM [3], Confident learning [4], SimiFeat [5],
(2) *Limited datasets*: the authors only experiment on a limited number of datasets, missing out on widely-used datasets for noisy learning e.g. Clothing1M (or its subset Clothing100K), or datasets of practical significance such as that of noisy chest X-rays (NoisyCXR) [6]. I would also be more interested in seeing transfer from CIFAR to NoisyCXR, or CIFAR to Clothing100K,
(3) *Multiple runs*: All results are reported over a single run, and hence it is not possible to compare the significance of differences,
(4) *Arbitrarily large noise rates*: The authors run experiments with 40% noise, which in my opinion, is unrealistic, and much higher than the noise rate of real world datasets,
(5) *Base classification models*: the authors only report their results on one base classification architecture, and not different architectures,
(6) *Methods to add label noise*: The study uses a limited number of synthetic label noise generation methods, when many other ways to inject label noise exist, such as class-dependent label noise.
Overall, I highly encourage the authors to follow and extend the evaluation protocol proposed in [1].
2. **Premise**: I am not convinced that training loss dynamics and confidence of trained models are indeed shared across datasets. Perhaps performing more experiments, on more datasets, as I suggested above might help demonstrate this. Also AUM and Confident learning use these very heuristics to identify labeling errors (and more), yet the authors fail to cite these papers.
3. **Clarity**: Some important details are missing from the paper. See Questions.
4. **Theory**: I don't think the theorem provided by the authors tells us anything different from what is already well known. Could the authors explain what is the new theoretical insight that they provide?

**References:**
[1] Goswami, M., Sanil, V., Choudhry, A., Srinivasan, A., Udompanyawit, C., & Dubrawski, A. (2023). AQuA: A Benchmarking Tool for Label Quality Assessment. arXiv preprint arXiv:2306.09467.

[2] Chong, Derek, Jenny Hong, and Christopher D. Manning. "Detecting Label Errors using Pre-Trained Language Models." arXiv preprint arXiv:2205.12702 (2022).

[3] Pleiss, G., Zhang, T., Elenberg, E., & Weinberger, K. Q. (2020). Identifying mislabeled data using the area under the margin ranking. Advances in Neural Information Processing Systems, 33, 17044-17056.

[4] Northcutt, C., Jiang, L., & Chuang, I. (2021). Confident learning: Estimating uncertainty in dataset labels. Journal of Artificial Intelligence Research, 70, 1373-1411.

[5] Zhu, Z., Dong, Z., & Liu, Y. (2022, June). Detecting corrupted labels without training a model to predict. In International conference on machine learning (pp. 27412-27427). PMLR.

[6] Bernhardt, M., Castro, D. C., Tanno, R., Schwaighofer, A., Tezcan, K. C., Monteiro, M., ... & Oktay, O. (2022). Active label cleaning for improved dataset quality under resource constraints. Nature communications, 13(1), 1161.

**Questions:**

1. Are you assuming that the source dataset it clean?
2. How is the source dataset corrupted?
3. What is the performance of the models without label cleaning?
4. Why do you only use Stochastic Gradient Descent for optimization? Can other optimization techniques such as Adam or AdamW not be used?
5. What is a theoretical insight that your theory provides that is not already well-known?
6. How would you compare your method with the use of foundation models to identify labeling errors (see [2])? The idea is simple: that losses of large pre-trained language models on text datasets are representative of labeling errors.

---

### Meta-Review · Area_Chair_Boo7 · 2023-12-12

**Metareview:**

This paper aims at a well-motivated need to address the challenge of noisy labels in supervised machine learning. It has been assessed by four knowledgeable reviewers, two of whom recommended straight rejection, one marginal rejection, and one - the briefest of the four - marginal acceptance. The authors have not engaged the reviewers in a discussion. The key issues brought up in the reviews included limited innovation, lacking experiments (insufficient scope of benchmark data, older alternatives used in comparisons), and an observation that the concept's dependance om meta-labels needed to learn how to identify noisy labels, is vulnerable to label noise as well. With those considerations, this paper in its current form is not fit for inclusion in the ICLR.

**Justification For Why Not Higher Score:**

This is a clear reject.

**Justification For Why Not Lower Score:**

N/A

---

### Decision · Program_Chairs · 2024-01-16

Reject